# Analysis of the *DYNC1H1* Gene Polymorphic Variants’ Association with ASD Occurrence and Clinical Phenotype of Affected Children

**DOI:** 10.3390/genes16050510

**Published:** 2025-04-28

**Authors:** Anna Balcerzyk-Matić, Tomasz Iwanicki, Alicja Jarosz, Tomasz Nowak, Ewa Emich-Widera, Beata Kazek, Agnieszka Kapinos-Gorczyca, Maciej Kapinos, Joanna Iwanicka, Katarzyna Gawron, Wirginia Likus, Paweł Niemiec

**Affiliations:** 1Department of Biochemistry and Medical Genetics, Faculty of Health Sciences in Katowice, Medical University of Silesia in Katowice, Medykow Street 18, 40-752 Katowice, Poland; tiwanicki@sum.edu.pl (T.I.); alicja.jarosz@sum.edu.pl (A.J.); tnowak@sum.edu.pl (T.N.); jiwanicka@sum.edu.pl (J.I.); pniemiec@sum.edu.pl (P.N.); 2Department of Pediatric Neurology, Faculty of Medical Science in Katowice, Medical University of Silesia in Katowice, Medykow Street 16, 40-752 Katowice, Poland; eemich-widera@sum.edu.pl; 3Child Development Support Center Persevere, Kępowa Street 56, 40-583 Katowice, Poland; beakazek@op.pl; 4Mental Health Center Feniks, Daily Ward for Children and Adolescents, Młyńska Street 8, 44-100 Gliwice, Poland; aga27.11@interia.eu (A.K.-G.); mkapi@o2.pl (M.K.); 5Department of Medical Microbiology, Faculty of Medical Sciences in Katowice, Medical University of Silesia in Katowice, Medykow Street 18, 40-752 Katowice, Poland; kgawron@sum.edu.pl; 6Department of Anatomy, Faculty of Health Sciences in Katowice, Medical University of Silesia in Katowice, Medykow Street 18, 40-752 Katowice, Poland; wlikus@sum.edu.pl

**Keywords:** autism spectrum disorder, polymorphism, genetic variants, dynein

## Abstract

Objectives: To analyze potential associations between three polymorphisms (rs3818188, rs941793, rs2403015) of the *DYNC1H1* gene and the occurrence of autism spectrum disorder as well as the clinical phenotype of affected individuals. Methods: This family-based study included 206 children diagnosed with ASD and 364 of their biological parents. To examine the potential association between three polymorphisms of the *DYNC1H1* gene and ASD occurrence, a transmission disequilibrium test was performed. Additionally, associations between the studied polymorphisms and the clinical phenotype of affected individuals were analyzed using the χ^2^ test. Results: None of the polymorphisms studied showed an association with ASD in the overall patient group. However, an association between the rs3818188 polymorphic variant and ASD was observed in a subgroup of girls, with the G allele being transmitted more than 2.5 times as frequently as the A allele. Moreover, several associations between the tested variants and features related to neuromotor development, communication, and social skills were observed in univariate analysis. However, after correction for multiple comparisons, only the association between the rs2403015 polymorphism and transient increase in muscle tone during infancy remained statistically significant. Conclusions: This study demonstrated an association between the rs3818188 polymorphism and ASD in a subgroup of girls. Additionally, the rs2403015 polymorphism was found to be associated with transient increase in muscle tone during infancy.

## 1. Introduction

Autism spectrum disorder (ASD) is a complex neurobiological condition in the etiology of which genetic factors play a significant role. In some cases, these are known factors determining the appearance of the clinical phenotype, such as certain chromosomal aberrations (e.g., causing Angelman syndrome) or single-gene diseases (e.g., Fragile X syndrome). Nevertheless, most cases of ASD are idiopathic with no apparent genetic cause. In such cases, the genetic basis of ASD may result from interactions between common genetic variants and environmental factors [1].

In order to identify such genetic variants, three polymorphisms of the *DYNC1H1* (dynein cytoplasmic 1 heavy chain 1) gene were tested. This gene encodes a protein that belongs to the dynein cytoplasmic heavy chain family. Dyneins are a group of microtubule-activated ATPases that act as molecular motors. They play a crucial role in intracellular transport, protein sorting, and organelle movement. This protein also fulfills neuron-specific functions, influencing gene expression, development, and regeneration; therefore, it is considered as a key protein for the proper functioning of the nervous system [2]. Previous reports showed associations between some rare variants of the *DYNC1H1* gene and several neurological pathologies, as well as neurodevelopmental and neuromuscular phenotypes [3,4]. Mutations in the *DYNC1H1* gene are also associated with autosomal dominant mental retardation-13 (MRD13), a form of intellectual disability linked to defects in neuronal migration, which result in malformations of the cerebral cortex. *DYNC1H1* mutations can severely compromise the core mechanochemical properties of human dynein. According to Hoang et al. [5], some of these mutations have strong effects on microtubule gliding by ensembles of dyneins, as well as on the processive movement of individual motor complexes. Others reduce the frequency of processive movement of individual microtubule-associated dyneins, shorten the travel distance of processive events, and lead to a significant slowing of these movements. These disruptions impair the transport of organelles, vesicles, and proteins along microtubules, which is essential for proper neuronal positioning and connectivity during brain development. As a result, cortical development can be altered, leading to malformations and contributing to neurodevelopmental disorders. Whole-genome sequencing studies have demonstrated de novo loss-of-function mutations in the *DYNC1H1* gene in some patients with ASD. A higher than expected mutation rate of this gene indicates it as an interesting candidate gene for studies on the pathogenesis of ASD [6]. According to the SFARI Gene database [7] and the Evaluation of Autism Gene Link Evidence (EAGLE) scoring system, the *DYNC1H1* gene was classified as a strong candidate gene for ASD. So far, common polymorphisms of the *DYNC1H1* gene have not been studied in patients with ASD, but it is suspected that they may influence gene expression and thus promote neurodevelopmental disorders.

In this study, three polymorphisms, i.e., rs941793, rs2403015, and rs3818188, located within various regions of the *DYNC1H1* gene, were analyzed. The rs941793 polymorphism is an intronic variant located in the region that encodes the motor domain and mutations associated with autism have been more frequently described in this domain [8]. This polymorphism is in linkage disequilibrium with many other polymorphisms, mainly intronic ones also located in this region of the *DYNC1H1* gene. The rs2403015 is located within a regulatory element of the *DYNC1H1* gene [9], whereas the rs3818188 is a synonymous variant found in the stem domain. Synonymous polymorphic variants do not change the amino acid sequence, and are therefore usually excluded from analyses; however, an increasing number of data indicate that they may affect mRNA expression and, consequently, protein conformation and activity [10,11]. In the literature, there are no data on the functional significance of the rs3818188 variant; however, several associations between this polymorphism and specific phenotypes have been described [12,13], which may suggest a potential functional relevance of the rs3818188. Initially, the criterion for polymorphism selection in this study was the minor allele frequency in the European population, not less than 0.20. However, due to the fact that most polymorphisms in the *DYNC1H1* gene have a lower frequency, and many are in linkage disequilibrium with the rs941793 polymorphism, eventually the rs2403015 polymorphic variant which showed a slightly lower frequency (0.18) was also included.

The aim of this study was to analyze the associations of three polymorphisms (rs3818188, rs941793, rs2403015) of the *DYNC1H1* gene with ASD occurrence and the clinical phenotype of affected individuals.

## 2. Materials and Methods

### 2.1. Study Design

We conducted a family-based study to investigate the role of genetic polymorphisms in the *DYNC1H1* gene in the development of autism spectrum disorder. The study included 206 children diagnosed with ASD and 364 of their biological parents. To determine the genetic association, we performed a transmission disequilibrium test (TDT) on three candidate polymorphisms in the *DYNC1H1* gene. The TDT assesses whether the alleles transmitted from parents to affected children are in equilibrium or exhibit a biased transmission pattern. We also analyzed the association between genetic variation and clinical characteristics of ASD using χ^2^ tests. The study was conducted in accordance with the Strengthening the Reporting of Observational studies in Epidemiology (STROBE) guidelines.

### 2.2. Materials

The study group included 570 individuals and consisted of 206 children diagnosed with ASD (average age 7.28 ± 2.72 years) and 364 of their biological parents. Boys constituted 78.6% (*n* = 162) of the study group, while girls made up 21.4% (*n* = 44). In total, 182 families were analyzed, 16 of them with two and four families with three affected children. Patients and their parents were recruited at the Department of Pediatric Neurology, John Paul II Upper Silesian Child Health Centre in Katowice, the Child Development Support Center in Katowice, and the Psychiatric Daily Ward for Children and Adolescents in Gliwice between 2016 and 2019. Based on the interview, direct psychiatric examination, and observation, the psychiatrist diagnosed the disorder according to the DSM-5 criteria. Each child also underwent an evaluation following the ADOS-2 protocol [14]. The results of both procedures, namely the psychiatric examination and the ADOS-2 protocol, confirmed the ASD diagnosis. Inclusion criteria were ASD diagnosis and an age of 3–12 years. Strict exclusion criteria were applied to obtain a homogeneous group of patients (which could be termed “nonsyndromic autism” or “pure autism group”), such as the presence of associated issues like epilepsy, intellectual disability, and other genetic and neurological diseases. The patient selection scheme is presented in Figure 1.

### 2.3. Genetic Analysis

DNA was isolated from human peripheral blood mononuclear cells (PBMCs) using the MasterPure™ Genomic DNA Purification Kit (Epicentre Technologies, Madison, USA). The *DYNC1H1* gene polymorphic variants (rs941793, rs2403015, and rs3818188) were genotyped using specifically designed TaqMan^®^ probes (Applied Biosystems, Foster City, CA, USA) and a Roche LightCycler^®^ 480II (F. Hoffmann-La Roche AG, Basel, Switzerland). Fifteen percent of DNA samples used were randomly re-genotyped for accuracy testing. The repeatability of the results in this study reached 100%. To determine whether the studied variants affect the *DYNC1H1* gene expression, we performed expression quantitative trait loci (eQTL) analysis. The data used for the analyses were obtained from the Genotype Tissue Expression (GTEx) Portal on 26 September 2024 [15]. The GTEx Project is a comprehensive public resource to study human gene expression and regulation, and its relationship to genetic variation across multiple diverse tissues and individuals. The project collected samples from up to 54 non-diseased tissue sites across nearly 1000 deceased adult individuals. The calculator makes it possible to compare gene expression between different polymorphic variants in various tissues.

### 2.4. Statistical Analysis

The Hardy–Weinberg equilibrium formula was used to calculate the expected values of the genotypes distribution. Then, using STATISTICA 13.3, we performed a χ^2^ test to determine whether the distribution of genotypes in our study group was consistent with the expected values. The transmission disequilibrium test was applied to analyze the potential association between specified polymorphisms and ASD. The TDT is based on analysis of the transmission of specific alleles from heterozygous parents to affected children (families referred to as informative trios). Families in which both parents are homozygous are not included in the analysis. If the disorder is not associated with given polymorphism, the expected frequencies of transmitted and non-transmitted alleles should be equal (50%). A significant deviation from this expected transmission rate suggests an association between the allele and the disease locus. Observed transmitted allele frequencies in the study were compared with the expected frequencies using the χ^2^ test. Associations between genetic variants and qualitative clinical features of children with ASD for dominant/recessive model were analyzed using the χ^2^ test, and if any subgroup consisted of fewer than 10 individuals, Yates’ correction was applied. For the additive model (genotypes), multi-way contingency tables and the χ^2^ test were used. The results were considered statistically significant at *p* < 0.05. In the case of multiple comparisons, the *p*-values were adjusted using the Hochberg correction method to decrease false positive results. The Hochberg correction, also called the step-up method, is based on a reverse scenario when the largest *p*-value is examined first. Once a significant *p*-value is identified, all the remaining smaller *p*-values would be declared significant [16]. Statistical analyses were performed using STATISTICA 13.3 and calculator for multiple comparisons on 26 September 2024 [https://multipletesting.com/analysis]. Haplotype analysis was conducted using Haploview 4.2 software [17]. Linkage disequilibrium (D′ and r^2^) was calculated using the standard algorithm by Gabriel et al. [18]. In this algorithm, the haplotype block creates a contiguous set of polymorphisms for which we do not observe recombination or it is very rare. Linked polymorphisms are considered to be those whose D′ value is within the range of 0.7–1. However, values below 0.9 may indicate recombination, which is why the mathematical value r^2^ is also used. This value should be greater than 0.7 (70%).

## 3. Results

### 3.1. Analysis of the Association Between DYNC1H1 Gene Polymorphism and ASD Occurrence

The distribution of genotypes and alleles for each polymorphism was consistent with the Hardy–Weinberg equilibrium. Haplotype analysis did not reveal significant associations between the tested polymorphisms (D′ = 1.0, R^2^ = 0.05 for rs3818188 and rs941793, D′ = 1.0, R^2^ = 0.74 for rs941793 and rs2403015, D′ = 1.0, R^2^ = 0.03 for rs3818188 and rs2403015). Genotype frequencies in the group of children and their parents are shown in Table 1. Among the analyzed polymorphisms, none showed an association with the ASD occurrence in the entire group of examined children (Table 2). The only observed trend was a more frequent transmission of the G allele of the rs3818188 polymorphic variant from heterozygous parents to their affected children. Nevertheless, an association between the rs3818188 polymorphic variant and ASD was observed exclusively in a subgroup of girls, where the G allele was transmitted more than 2.5 times as often as the A allele (Table 3).

### 3.2. Analysis of the Association Between DYNC1H1 Gene Polymorphism and Clinical Phenotype

Considering that the *DYNC1H1* gene is involved in nervous system development, and that mutations in this gene are associated with neurodevelopmental and neuromuscular phenotypes, as well as intellectual disability, we further investigated potential correlations between genotypes and phenotypes of children’s neuromotor development. Furthermore, we explored associations between genotypes and phenotypes related to the core domains affected in ASD, namely communication, social interaction, and behavior. The clinical characteristics of examined patients, based on an interview regarding infancy and the present period, are summarized in Table 4. The highest percentage of patients were characterized by sensory integration (SI) impairments, compulsive behavior, and limited eye contact. Several associations of the tested polymorphisms with communication and social-related features were observed in a univariate analysis (Table 5). Carriers of the A allele of the rs3818188 polymorphic variant were more often characterized by the ability to engage in pretend play compared to individuals with the GG genotype. In the case of the rs941793 polymorphism, carriers of the G allele performed the ”bye-bye” gesture less often in comparison to AA homozygotes. Additionally, in carriers of the G allele of the rs2403015 polymorphic variant transient increase in muscle tone during infancy was observed more frequently compared to patients with the AA genotype. We also observed an association of the rs2403015 polymorphic variant genotype with hearing impairments in the additive model. Overall, we analyzed associations of 16 phenotypic traits with three polymorphisms employing two models (additive and dominant/recessive); thus, in total 96 tests were performed, which undoubtedly increased the probability of false positive associations of non-negligible proportion. To overcome the issue of multiple hypothesis testing, we used the Hochberg correction for multiple comparisons. This correction showed an association between the G allele carrier-state of the rs2403015 polymorphism and transient increase in muscle tone during infancy as one of significance.

### 3.3. In Silico Analysis of the Influence of Polymorphic Variants on Gene Expression

To verify whether analyzed variants of the *DYNC1H1* gene may affect its expression, an in silico analysis of expression quantitative trait loci (eQTL) was conducted. We demonstrated that the G allele was associated with decreased gene expression in skeletal muscle (*p* value = 0.000034, *p* value threshold = 0.00016, NES = −0.078, T statistic = −4.2) (Figure 2 from https://gtexportal.org/home/ on 27 September 2024), whereas no correlation was found between any of the analyzed variants and gene expression in brain tissue.

## 4. Discussion

The association of *DYNC1H1* gene polymorphic variants and ASD occurrence was not found in the entire group of patients enrolled in this study; however, it was demonstrated for the rs3818188 polymorphism in a subgroup of examined girls with ASD. The A (minor) allele was transmitted less frequently to affected girls by heterozygous parents compared to the G allele, which may suggest a protective role of the A allele. This polymorphism is located in the region encoding the stem domain, where mutations associated with neuromotor disorders have previously been reported [8]. The analyzed variant represents a synonymous mutation, which does not alter the primary structure of the protein and may appear to have no functional significance. However, increasing evidence indicates that synonymous mutations can influence mRNA expression, as well as protein conformation and activity, potentially leading to disease manifestation. An example of such a variant is the c.261C>T mutation in the *DYNC1H1* gene which increases the sensitivity of the protein to protease digestion and thermal degradation, potentially affecting its structure [11]. In the literature, there are no data on the functional significance of the rs3818188 variant; however, associations between this polymorphism and specific phenotypes have been described, which may suggest a potential functional relevance for rs3818188. Liu et al. [12] identified it as a risk variant for chronic obstructive pulmonary disease, and Huang et al. [13] observed an association between this polymorphism and the efficacy of glucocorticoids in the treatment of systemic lupus erythematosus in males.

There are not many studies that could help explain why differences in allele transmission have been observed only in girls. It can be hypothesized that this may be related to the influence of sex hormones on the development and function of the central nervous system. Since the DYNC1H1 protein also performs neuron-specific functions, influencing gene expression, development, and regeneration, it is possible that it interacts in some way with sex hormones in these processes. According to Asemota et al. [19], the upstream DNA element of the *DYNC1H1* gene contains androgen response elements. Additionally, the hallmark early estrogen response signaling pathway has been significantly correlated with *DYNC1H1* expression [20]. Nevertheless, the mechanism of action of the *DYNC1H1* gene, its role in the proper development of the nervous system, and its interactions with other signaling pathways are still not fully understood.

In previous studies, mutations present in the motor domain of the DYNC1H1 protein have been more frequently associated with ASD, while the variants associated with neuromuscular symptoms were clustered in the stem domain [8]. In our study, the only polymorphism located in the motor domain coding region was rs941793, but we did not observe its association with ASD. This intronic polymorphism is linked to many common polymorphisms of the *DYNC1H1* gene, which were therefore not analyzed in this study. Univariate analysis of the impact of rs941793 on neuromotor development and ASD-related phenotypic features showed that G allele carriers were less likely to perform the bye-bye gesture; however, after Hochberg correction for multiple comparisons, this difference was not statistically significant, which suggests that it could be a false positive observation.

An interesting observation was made in the case of the rs2403015 polymorphic variant. The G allele was found to be associated with transient increase in muscle tone during infancy, both in univariate analysis and after Hochberg correction. This polymorphism is located within a regulatory element, which may affect gene expression by regulating transcription, splicing, or mRNA stability [9]. The eQTL in silico analysis showed an impact of this polymorphic variant on *DYNC1H1* gene expression in skeletal muscle, but not in brain tissue. Univariate analysis also showed the association of the rs2403015 polymorphism with hearing impairment, as well as the association of the rs3818188 polymorphism with the ability to engage in pretend play, but these observations were not confirmed after correction for multiple comparisons, raising concerns about their reliability.

In conclusion, this study showed an association between the rs3818188 polymorphism and ASD in a subgroup of girls, with the G allele being transmitted more than 2.5 times as frequently as the A allele. Additionally, the rs2403015 polymorphism was found to be associated with transient increase in muscle tone during infancy of affected children. The results obtained in this study may be helpful in determining the direction of further research aimed at identifying genetic markers of ASD.

### Study Limitations

Although the entire study group was quite large, some of the results, including the most significant one regarding the association between the rs3818188 polymorphism and ASD, were obtained by analyzing a subgroup of girls, which significantly reduced the size of the study group. Moreover, the TDT test can only be conducted in the case of informative trios (where at least one parent is heterozygous), further limiting the number of patients. Therefore, these results should be confirmed in a larger cohort. The frequency of the rarer allele of the selected polymorphisms was also relatively low, leading to a smaller number of heterozygotes. This, in turn, was due to the fact that a large number of common polymorphisms in the *DYNC1H1* gene are linked to the rs941793 polymorphism included in our analysis, which is why we decided not to include them in our analysis. However, we attempted to select polymorphisms located in different regions of the gene and those that may have potential functional significance. Unfortunately, we were not able to provide experimental data confirming the effect of the studied polymorphisms at the RNA and/or protein level, which would have significantly increased the impact and scientific value of this study.

In conclusion, although some associations between the analyzed *DYNC1H1* gene polymorphisms and ASD or clinical phenotypes were identified, the results require confirmation in a larger cohort.

## Figures and Tables

**Figure 1 genes-16-00510-f001:**
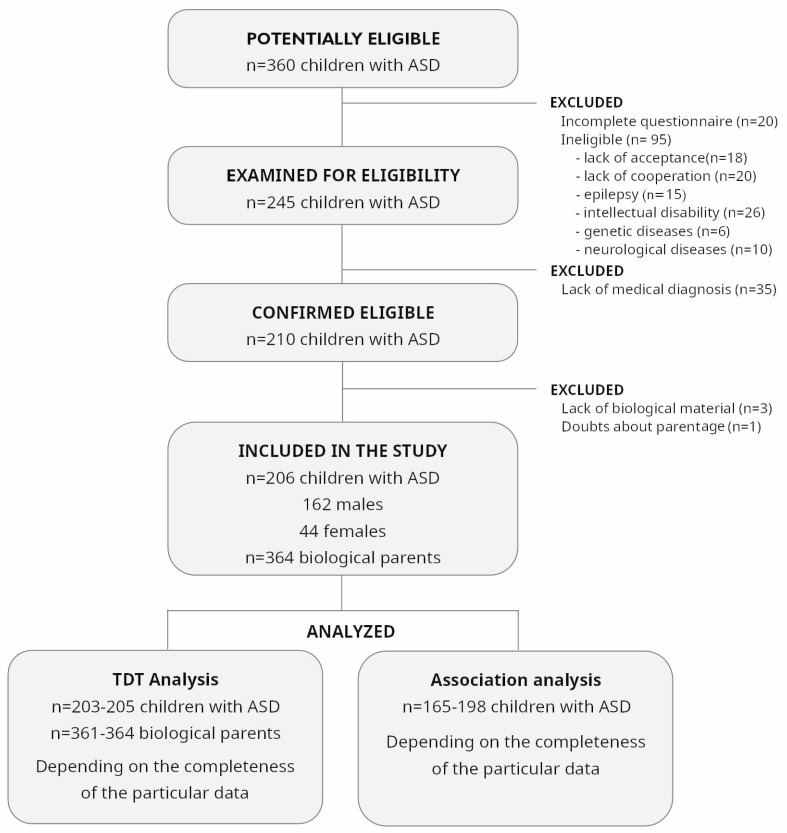
The flow chart presenting the criteria of patients’ recruitment for the study (TDT—transmission disequilibrium test).

**Figure 2 genes-16-00510-f002:**
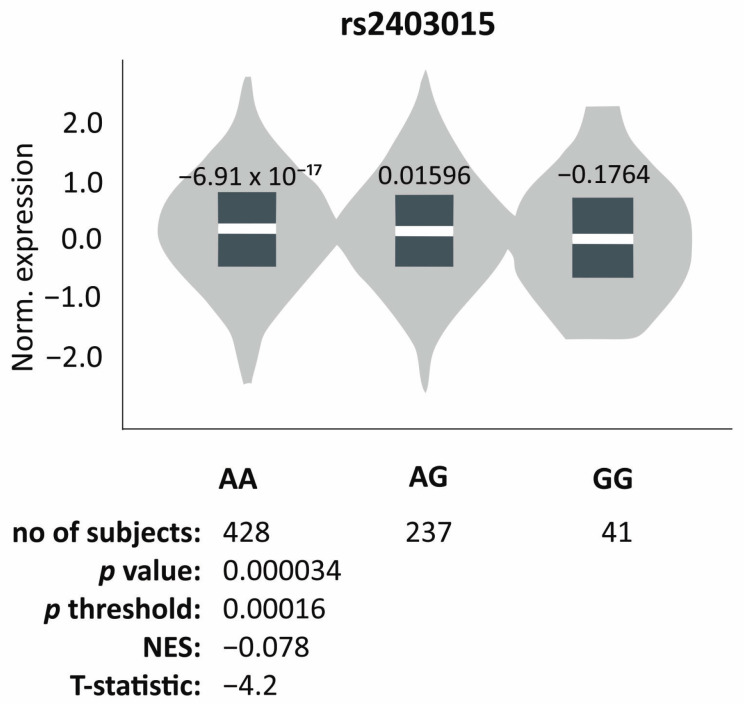
Gene expression of *DYNC1H1* in skeletal muscle tissue dependently on the genotype of the rs2403015 polymorphism. Based on GTEx Portal [15].

**Table 1 genes-16-00510-t001:** Distribution of genotypes and alleles in children and their biological parents.

Genotype		Genotype Distribution, n (%)		
		Children	Mothers	Fathers
rs3818188	n	205	205	205
GG		132 (64.4)	135 (65.8)	118 (57.6)
GA		66 (32.2)	60 (29.3)	79 (38.5)
AA		7 (3.4)	10 (4.9)	8 (3.9)
rs941793	n	205	205	205
AA		141 (68.8)	137 (66.8)	142 (69.3)
AG		55 (26.8)	59 (28.8)	54 (26.3)
GG		9 (4.4)	9 (4.4)	9 (4.4)
rs2403015	n	203	202	203
AA		153 (75.4)	144 (71.3)	154 (75.9)
AG		44 (21.7)	55 (27.2)	42 (20.7)
GG		6 (2.9)	3 (1.5)	7 (3.4)

**Table 2 genes-16-00510-t002:** Transmission disequilibrium test for the entire group (n = 206).

Allele	Transmitted n (%)	Not Transmitted n (%)	χ^2^; *p*
rs3818188	n informative trios = 115		
G A	77 (55.4) 62 (44.6)	62 (44.6) 77 (55.4)	1.619, 0.203
rs941793	n informative trios = 99		
A G	57 (50.4) 56 (49.6)	56 (49.6) 57 (50.4)	0.009, 0.925
rs2403015	n informative trios = 86		
A G	50 (52.1) 46 (47.9)	46 (47.9) 50 (52.1)	0.167, 0.683

**Table 3 genes-16-00510-t003:** Transmission disequilibrium test for rs3818188 polymorphism for male (n = 162) and female (n = 44).

Allele	Transmitted n (%)	Not Transmitted n (%)	χ^2^; *p*
rs3818188	Boys n informative trios = 86		
G A	54 (50.5) 53 (49.5)	53 (49.5) 54 (50.5)	0.019; 0.891
rs3818188	Girls n informative trios = 29		
G A	23 (71.9) 9 (28.1)	9 (28.1) 23 (71.9)	6.125; 0.013 *

* Statistically significant difference.

**Table 4 genes-16-00510-t004:** Clinical characteristics of the study group.

Characteristics	n (%)	n Available Data *
Abnormal motor development	79 (41.6)	190
Transient increase in muscle tone (infant period)	37 (19.3)	192
Decrease in muscle tone	67 (35.3)	190
Asymmetric position	45 (23.2)	194
SI impairments	172 (89.6)	192
Need for rehabilitation	64 (32.3)	198
Hearing impairments	32 (16.6)	193
Vision impairments	54 (28.1)	192
Regression in communication	77 (39.5)	195
Impairment in eye contact	135 (74.6)	181
Bye-bye gesture	119 (68.0)	175
Reaction to name	131 (67.5)	194
Pretend play	59 (30.4)	194
Compulsive, ritualistic behaviour	141 (75.0)	188
Self-aggressive behaviour	78 (39.8)	196

* Differences in the number of children in specific groups result from the lack of some clinical data.

**Table 5 genes-16-00510-t005:** Associations of studied polymorphisms with clinical phenotypes of affected children.

**Genotype Frequency of rs3818188 Polymorphism and Skill of Pretend Play (Recessive/Dominant Model)**				
	Yes (n, %)	No (n, %)		χ^2^; *p*
AA + GA GG	28 (47.5) 31 (52.5)	43 (31.9) 92 (68.1)	vs. GG	4.31; 0.038 ^1^
**Genotype frequency of rs941793 polymorphism and bye-bye gesture** (**recessive/dominant model)**				
	Yes (n, %)	No (n, %)		χ^2^; *p*
GG + AG AA	29 (24.4) 90 (75.6)	22 (39.3) 34 (60.7)	vs. AA	4.10; 0.043 ^1^
**Genotype frequency of rs2403015 polymorphism and** transient **increase in muscle tone during infancy** (**recessive/dominant model)**				
	Yes (n, %)	No (n, %)		χ^2^; *p*
GG + AG AA	14 (37.8) 23 (62.2)	30 (19.4) 125 (80.6)	vs. AA	5.78; 0.016 ^1,2^
**Genotype frequency of rs2403015 polymorphism and hearing impairments (additive model)**				
	Yes (n, %)	No (n, %)		χ^2^*; p*
AA AG GG	27 (84.4) 3 (9.4) 2 (6.2)	121 (75.2) 38 (23.6) 2 (1.2)		6.07; 0.048 ^1^

^1^ Statistically significant differences in univariate analysis. ^2^ Statistically significant differences after Hochberg correction for multiple comparisons. Differences in the number of children in specific groups resulted from the lack of complete clinical data.

## Data Availability

The data that support the findings of this study are available from the corresponding author, [A.B.-M.], upon reasonable request.

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
