# Peer review of "Analysis of the DYNC1H1 Gene Polymorphic Variants’ Association with ASD Occurrence and Clinical Phenotype of Affected Children"

_genes, 2025, doi:10.3390/genes16050510_

Round 1
Reviewer 1 Report
Comments and Suggestions for Authors
The article titled "Analysis of the DYNC1H1 gene polymorphic variants association with ASD occurrence and clinical phenotype of affected children" investigates the potential link between three specific polymorphisms—rs3818188, rs941793, and rs2403015—in the DYNC1H1 gene and both the occurrence and clinical features of Autism Spectrum Disorder (ASD) in children. The study addresses a timely and clinically relevant question regarding the genetic underpinnings of ASD through a well-rationalized candidate gene approach.
Key Findings:
- rs3818188 showed a significant association with ASD in girls, suggesting a possible sex-specific genetic effect.
- rs2403015 was significantly associated with transient increased muscle tone in infancy, highlighting potential neuromotor implications of this variant.
Strengths:
- The methodology appears robust, with thoughtful statistical analysis and appropriate consideration of clinical sub phenotypes.
- The authors provide plausible biological interpretations, discussing mechanisms such as the influence of synonymous variants on mRNA stability or translation.
- The inclusion of in silico eQTL analysis supports initial functional hypotheses.
Limitations:
- Lack of experimental validation: The study does not include in vitro or in vivo experiments to confirm whether these polymorphisms affect DYNC1H1 expression or function at the RNA/protein level.
- Tissue specificity of eQTL analysis: The functional association for rs2403015 was observed in skeletal muscle, not neural tissue, limiting direct relevance to neurodevelopmental outcomes. eQTL data from fetal or developing brain tissue would be more informative.
- Sex distribution imbalance: The cohort's ~4:1 male-to-female ratio mirrors ASD prevalence but reduces statistical power for sex-stratified analyses, especially for females. The findings require replication in larger, independent cohorts, ideally with a more balanced sex distribution to validate sex-specific effects.
Author Response
Dear Reviewer,
Thank you very much for reviewing our paper and for your valuable comments. Below, we provide responses to each of them. The changes in the manuscript are highlighted in red.
1) Lack of experimental validation: The study does not include in vitro or in vivo experiments to confirm whether these polymorphisms affect DYNC1H1 expression or function at the RNA/protein level.
Unfortunately, during our study, there was no possibility to conduct this type of analysis. Of course, providing experimental data confirming the effect of the studied polymorphisms at the RNA and/or protein level would significantly contribute to research on DYNC1H gene polymorphisms in ASD. Therefore, this is an interesting research direction to consider in the future. We have added this information to the Limitations section.
2) Tissue specificity of eQTL analysis: The functional association for rs2403015 was observed in skeletal muscle, not neural tissue, limiting direct relevance to neurodevelopmental outcomes. eQTL data from fetal or developing brain tissue would be more informative.
The tool we used to perform eQTL analyses unfortunately does not allow analyses in tissues from fetal or developing brain. However, we performed analyses of the association of rs2403015 with gene expression in the following brain tissues: amygdala, anterior cingulate cortex, caudate basal ganglia, cerebellar hemisphere, cerebellum, cortex, frontal cortex, hippocampus, hypothalamus, nucleus accumbens of the basal ganglia, putamen basal ganglia, cervical spinal cord, substantia nigra, mammary gland tissue. Unfortunately, none of these analyses revealed a significant association. Of course, experimental validation of the obtained results in the future would provide greater insight into the actual effect of rs2403015 on gene expression.
3) Sex distribution imbalance: The cohort's ~4:1 male-to-female ratio mirrors ASD prevalence but reduces statistical power for sex-stratified analyses, especially for females. The findings require replication in larger, independent cohorts, ideally with a more balanced sex distribution to validate sex-specific effects.
We agree that performing analyses in larger and more balanced cohorts would contribute to confirming our results (what is underlined in Limitations section). Although it is not currently possible for us to perform such analyses, this is an important research direction related to the analysis of the DYNC1H1 gene to consider in the future.
Reviewer 2 Report
Comments and Suggestions for Authors
Comments to the Auhtors
Comments to the Auhtors
This study suggest the association of the rs3818188 polymorphism with ASD in a subgroup of girls while the rs2403015 polymorphism is associated with transient increase in muscle tone during infant period
-Since the associations of some rare variants of the DYNC1H1 gene with several neurodevelopmental and neuromuscular isorderrs (cyte 3, 4), please, indicate the affected signalling pathways by these mutations in the DYNC1H1 gene , spetially in the cerebral cortex..
-Shall you explain the rs941793, rs2403015 and rs3818188 the evaluation of these DYNC1H1 gene polymorphims since the rs941793 polymorphism is an intronic variant althogh seem to play a role in autism?
-Shall you explain why these polymorphisms may affect mRNA expression and, consequently, protein conformation and activity¿
-Although the criterion for polymorphisms tested here the minor allele frequency in the European population, not less than 0.20. However, most polymorphisms in the DYNC1H1 gene have a lower frequency, and many are in linkage disequilibrium with the s941793 polymor-84 phism, eventually the rs2403015 polymorphic variant, which showed a slightly lower frquency (0.18) was also included. Shall you justify in detail the election criteria for these evaluated polimorphisms in your study?
Material and methods
-Describe the transmission disequilibrium test with detail in your material and method section.In my opinion the association between the studied polymorphisms and the clinical ASD phenotype of affected individuals should be analyzed by r Spearman or Pearson study. However, Kruskal-Wallis test evaluated a posible signifficant difference between groups in case of parametric data. Shall you clarify this point in your statistical analysis?
-The study design is not clear for me. For example, these authors indicate ¨ Study Design 92
-Explain with details the followed procedure used for ADS diagnosis by a psychiatry. ¨ The 106 ASD was diagnosed by a psychiatrist on the basis of the Autism Diagnosis Observation 107 Schedule (ADOS-2) protocol as the gold standard observation tool [11]¨
-Additionally, explain the difference between a homogeneous group of patients (which could be termed "nonsyndromic autism" or "pure autism group"), such as the presence of associated problems such as epilepsy, intellectual disability etc.
-Give more information about the followed procedure in line 122. They textually indicate ¨we performed expression quantitative trait loci (eQTL) analysis. The data used for the analyses were obtained 123 from the GTEx Portal on 26/09/24 [12]¨, including the χ² test for The Hardy-Weinberg equilibrium analysis in your study. Explain the statistical meaning of linkage disequilibrium (D′ and R²) for a general audience by the standard algorithm (Gabriel et al. [15].
-Since the rs3818188 polymorphic variant association with ASD was observed exclusively in a subgroup of girls, where the G allele was transmitted more than 2.5 times as often as the A allele (Table 3, please, indicate the clnical consequences in ADS of this variant in girls in the context of ASD patholoogy.
.Explain how the associations of the tested polymorphisms with communication and social-related features were measured in the univariate analysis (Table 5).
-Shall you justify why carriers of 167 the A allele of the rs3818188 polymorphic variant were more often characterized by the 168 ability to engage in pretend play compared to individuals with the GG genotype
-Please, explain how has been demostrated that the G allele was associated with decreased gene expression in skeletal muscle (p value = 0.000034, p value threshold = 0.00016, NES = -0.078, T statistic = -4.2, Figure 2 from https://gtexportal.org/home/), while any correlation of all analyzed variants with gene expression was found in brain tissue. Have you done postmotem studies with samples from brain bank tissues?
-Also indicate how have you calculate data from Figure 2 follwing the Based on GTEx Portal [12]. Gene expression of DYNC1H1 in skeletal muscle tissue dependently on the genotype of 194 the rs2403015 polymorphism.
Have you done any kind of statistical analysis able to differentiate a possible difference between genotypes of figure 3 about Genotype distribution, with n (%) . In addition, explain the Hochberg correction followed for multiple comparisons in table-4.
The organization of data allow to follow the study and the discussion cover all described ppolymorphisms. The inclusion of limitations define possible flaws of this study but it is great to include them. However, I miss the final conclussion of the study. Please, add a short final conclussion after all limitations, which are really important in this study.
My Decisiom is minnor revision
Comments on the Quality of English LanguageThe english style can be improved in this manuscript.
Author Response
Dear Reviewer,
Thank you very much for reviewing our paper and for your valuable comments. Below, we provide responses to each of them. The changes in the manuscript are highlighted in red. We have also carried out a language revision of the manuscript; however, we did not track the changes to maintain the clarity of the text.
1) Since the associations of some rare variants of the DYNC1H1 gene with several neurodevelopmental and neuromuscular isorderrs (cyte 3, 4), please, indicate the affected signalling pathways by these mutations in the DYNC1H1 gene , spetially in the cerebral cortex.
In the introduction, we added a few sentences about the signaling pathways that may be affected by mutations in the DYNC1H1 gene, as well as how these mutations may influence cortical development.
2) Shall you explain the rs941793, rs2403015 and rs3818188 the evaluation of these DYNC1H1 gene polymorphims since the rs941793 polymorphism is an intronic variant althogh seem to play a role in autism?
-Shall you explain why these polymorphisms may affect mRNA expression and, consequently, protein conformation and activity?
Although introns are non-coding sequences, their polymorphisms can influence gene expression in various ways. Introns may contain enhancers or silencers that modulate gene expression. These polymorphisms can also affect the process of alternative exon splicing, as well as RNA stability and transport.
3) Although the criterion for polymorphisms tested here the minor allele frequency in the European population, not less than 0.20. However, most polymorphisms in the DYNC1H1 gene have a lower frequency, and many are in linkage disequilibrium with the s941793 polymor-84 phism, eventually the rs2403015 polymorphic variant, which showed a slightly lower frquency (0.18) was also included. Shall you justify in detail the election criteria for these evaluated polimorphisms in your study?
When selecting polymorphisms for analysis, we were looking for:
- polymorphisms with functional significance,
- polymorphisms associated with various phenotypes,
- polymorphisms with a sufficiently high frequency in the population to ensure adequate statistical power.
However, it turned out that although the DYNC1H1 gene is a strong candidate for ASD, its polymorphisms are poorly studied in terms of functionality, and most of them have very low frequencies. Following careful evaluation, we selected three polymorphisms that most closely aligned with the initial criteria.
The rs2403015 polymorphism is located within a regulatory region (enhancer); therefore, it may have a significant impact on the regulation of gene expression by affecting enhancer activity and its ability to bind transcription factors. The eQTL in silico analysis showed also an impact of this polymorphic variant on DYNC1H1 gene expression in skeletal muscle. For this reason, we found this polymorphism interesting and included it in our analysis, even though its frequency was slightly lower than we had initially assumed.
In the literature, there is no data on the functional significance of the rs3818188 variant; however, several associations between this polymorphism and specific phenotypes have been described [12, 13]. This may suggest a potential functional relevance of rs3818188 (We removed one piece of information regarding this polymorphism from the manuscript, as we realized it was based on a misinterpretation of data from the Ensembl database).
Since previous studies have shown that mutations located in the motor domain of the DYNC1H1 protein were more frequently associated with ASD, we also wanted to analyze a polymorphism from the region of the gene encoding this domain. Therefore, we selected rs941793, which is in linkage with several other polymorphisms within this domain.
Material and methods
4) Describe the transmission disequilibrium test with detail in your material and method section. In my opinion the association between the studied polymorphisms and the clinical ASD phenotype of affected individuals should be analyzed by r Spearman or Pearson study. However, Kruskal-Wallis test evaluated a posible signifficant difference between groups in case of parametric data. Shall you clarify this point in your statistical analysis?
After a more in-depth analysis, we must acknowledge that the Kruskal-Wallis test was not quite suitable for our data. However, Spearman’s and Pearson’s correlation tests, designed for continuous variables, were also not appropriate, as all of our clinical data were of a categorical nature (1, 0). Therefore, we decided to use multi-way contingency tables along with the χ² test. This choice did not significantly impact the results of our analysis. Only a minor revision was made to Table 5.
Some information about the transmission disequilibrium test was added.
5) The study design is not clear for me. For example, these authors indicate ¨ Study Design 92
Study design section has been rewritten. We hope that it will be clearer now.
6) Explain with details the followed procedure used for ADS diagnosis by a psychiatry. ¨ The 106 ASD was diagnosed by a psychiatrist on the basis of the Autism Diagnosis Observation 107 Schedule (ADOS-2) protocol as the gold standard observation tool [11]¨
Thank you for pointing out the section in which the psychiatric diagnostic procedure is described. The submitted version is somewhat awkward in this regard. It seems that the following description would be more appropriate:
"Based on the interview, direct psychiatric examination, and observation, the psychiatrist diagnosed the disorder according to the DSM-5 criteria. Each child also underwent an evaluation following the ADOS-2 protocol. The results of both procedures, namely the psychiatric examination and the ADOS-2 protocol, confirmed the ASD diagnosis."
7) Additionally, explain the difference between a homogeneous group of patients (which could be termed "nonsyndromic autism" or "pure autism group"), such as the presence of associated problems such as epilepsy, intellectual disability etc.
Nonsyndromic autism and pure autism are treated as synonyms.
In an effort to obtain as homogeneous a group of autistic patients as possible, the authors included only children without comorbidities that could potentially affect the results and their interpretation. Therefore, the coexistence of autism with epilepsy, intellectual disability, and other neurological and genetic disorders was an exclusion criterion.
8) Give more information about the followed procedure in line 122. They textually indicate ¨we performed expression quantitative trait loci (eQTL) analysis. The data used for the analyses were obtained 123 from the GTEx Portal on 26/09/24 [12]¨, including the χ² test for The Hardy-Weinberg equilibrium analysis in your study. Explain the statistical meaning of linkage disequilibrium (D′ and R²) for a general audience by the standard algorithm (Gabriel et al. [15].
Additional information has been added.
9) Since the rs3818188 polymorphic variant association with ASD was observed exclusively in a subgroup of girls, where the G allele was transmitted more than 2.5 times as often as the A allele (Table 3, please, indicate the clnical consequences in ADS of this variant in girls in the context of ASD patholoogy.
Due to the very limited literature data regarding the functional significance of the rs3818188 polymorphism and potential sex-dependent differences in the activity of the DYNC1H1 protein in the developing brain, we can only suggest that the observed association may be related to the influence of sex hormones on the development and function of the central nervous system, as already discussed.
10) Explain how the associations of the tested polymorphisms with communication and social-related features were measured in the univariate analysis (Table 5).
In the univariate analysis, we examined the associations between individual polymorphisms and all the traits listed in Table 4. In terms of communication, these included, for example, regression in communication and performing the 'bye-bye' gesture (both presented as qualitative variables: 1 – regression, 0 – no regression). Traits related to social development included response to name and pretend play. The analyses were conducted using the χ² test, with Yates' correction applied when any subgroup consisted of fewer than 10 individuals.
11) Shall you justify why carriers of 167 the A allele of the rs3818188 polymorphic variant were more often characterized by the 168 ability to engage in pretend play compared to individuals with the GG genotype
The association of the rs3818188 polymorphism with the ability to engage in pretend play, similar to that of rs941793 with performing the 'bye-bye' gesture, was not statistically significant after applying the Hochberg correction for multiple comparisons. Therefore, we considered these to be false-positive results and did not attempt to interpret them.
12) Please, explain how has been demostrated that the G allele was associated with decreased gene expression in skeletal muscle (p value = 0.000034, p value threshold = 0.00016, NES = -0.078, T statistic = -4.2, Figure 2 from https://gtexportal.org/home/), while any correlation of all analyzed variants with gene expression was found in brain tissue. Have you done postmotem studies with samples from brain bank tissues?
-Also indicate how have you calculate data from Figure 2 follwing the Based on GTEx Portal [12]. Gene expression of DYNC1H1 in skeletal muscle tissue dependently on the genotype of 194 the rs2403015 polymorphism.
No, we haven’t done postmortem studies with samples from brain bank tissues. The lack of experimental validation was added to Limitations. We’ve only done analysis using the GTEx eQTL Calculator. The Adult Genotype Tissue Expression (GTEx) Project is a comprehensive public resource to study human gene expression and regulation, and its relationship to genetic variation across multiple diverse tissues and individuals. The project collected samples from up to 54 non-diseased tissue sites across nearly 1,000 deceased individuals. The calculator makes it possible to compare gene expression between different polymorphic variants in various tissues. Some additional information was added to Materials and Methods section.
13) Have you done any kind of statistical analysis able to differentiate a possible difference between genotypes of figure 3 about Genotype distribution, with n (%) . In addition, explain the Hochberg correction followed for multiple comparisons in table-4.
We are not sure what did you mean by “statistical analysis able to differentiate a possible difference between genotypes of figure 3”, also we don’t have a figure 3. Can you clarify what you meant?
Regarding the Hochberg correction: conducting multiple comparisons increases the likelihood that a non-negligible proportion of associations will be false positives, thereby clouding true discoveries. Several strategies exist to overcome this problem, and based on the analysis of the work by Menyhart et al. [16] we chose the Hochberg correction. This correction, also called the step-up method, is based on a reverse scenario where the largest p-value is examined first. Once a significant p-value is identified, all remaining smaller p-values are declared significant. We have added additional information to the Materials and Methods section.
14) The organization of data allow to follow the study and the discussion cover all described polymorphisms. The inclusion of limitations define possible flaws of this study but it is great to include them. However, I miss the final conclussion of the study. Please, add a short final conclussion after all limitations, which are really important in this study.
We’ve added a short final conclusion after Limitations section.